# Codon Directional Asymmetry Suggests Swapped Prebiotic 1st and 2nd Codon Positions

**DOI:** 10.3390/ijms21010347

**Published:** 2020-01-05

**Authors:** Hervé Seligmann, Jacques Demongeot

**Affiliations:** 1The National Natural History Collections, The Hebrew University of Jerusalem, 91404 Jerusalem, Israel; 2Faculty of Medicine, Université Grenoble Alpes, Laboratory AGEIS EA 7407, Team Tools for e-Gnosis Medical, F-38700 La Tronche, France; jacques.demongeot@yahoo.fr

**Keywords:** anticodon, amino-imino tautomer, keto-enol tautomer, systematic nucleotide exchange, systematic nucleotide deletion, swinger RNA, bijective transformation

## Abstract

**Background**: Codon directional asymmetry (CDA) classifies the 64 codons into palindromes (XYX, CDA = 0), and 5′- and 3′-dominant (YXX and XXY, CDA < 0 and CDA > 0, respectively). Previously, CDA was defined by the purine/pyrimidine divide (A,G/C,T), where X is either a purine or a pyrimidine. For the remaining codons with undefined CDA, CDA was defined by the 5′ or 3′ nucleotide complementary to Y. This CDA correlates with cognate amino acid tRNA synthetase classes, antiparallel beta sheet conformation index and the evolutionary order defined by the self-referential genetic code evolution model (CDA < 0: class I, high beta sheet index, late genetic code inclusion). **Methods**: We explore associations of CDAs defined by nucleotide classifications according to complementarity strengths (A:T, weak; C:G, strong) and keto-enol/amino-imino groupings (G,T/A,C), also after swapping 1st and 2nd codon positions with amino acid physicochemical and structural properties. **Results**: Here, analyses show that for the eight codons whose purine/pyrimidine-based CDA requires using the rule of complementarity with the midposition, using weak interactions to define CDA instead of complementarity increases associations with tRNA synthetase classes, antiparallel beta sheet index and genetic code evolutionary order. CDA defined by keto-enol/amino-imino groups, 1st and 2nd codon positions swapped, correlates with amino acid parallel beta sheet formation indices and Doolittle’s hydropathicities. **Conclusions**: Results suggest (a) prebiotic swaps from N2N1N3 to N1N2N3 codon structures, (b) that tRNA-mediated translation replaced direct codon-amino acid interactions, and (c) links between codon structures and cognate amino acid properties.

## 1. Introduction

Symmetry and symmetry breaking is one of the most powerful tools at our disposal for analyzing and understanding natural phenomena because it has clear mathematical properties, defines an absolute optimum state (perfect symmetry) and enables estimating discrepancies from it in terms of extents of asymmetry. This concept can be applied to many objects under study, such as molecules [1], minerals [2] or whole organisms [3,4,5,6,7,8,9,10,11]. It was also applied to the genetic code table [12], and more recently, to the structure of the genetic code’s codons, in relation to encoded amino acids [13].

### 1.1. Defining Codon Directional Asymmetry

Codon directional asymmetry (CDA) [13] defines side-dominance for codons based on nucleotides at codon 5′ vs. 3′ extremities (1st vs. 3rd codon positions), in comparison to the nucleotide at midposition (2nd codon position). Codons are considered palindromic, CDA = 0, when the same nucleotide occurs at first and last codon positions. Hence, codons with the same nucleotide at all three codon positions have CDA = 0 (AAA, CCC, GGG, TTT). Codons with two types of nucleotides, if the same nucleotide occurs at first and last positions, are also palindromic, and CDA = 0: CTC, ATA, GTG, TCT, ACA, GCG, TAT, CAC, GAG, TGT, CGC, AGA. Hence, 16 among 64 codons have CDA = 0. Twelve codons have the same nucleotide at first and second positions and a different nucleotide at third position. They have 3′-side dominance, CDA > 0: TTC, TTA, TTG, CCT, CCA, CCG, AAT, AAC, AAG, GGT, GGC and GGA. Codons with the same nucleotide at second and third positions, and a different nucleotide at first position, have CDA < 0: CTT, ATT, GTT, TCC, ACC, GCC, TAA, CAA, GAA, TGG, CGG, and AGG. The decision to define 3′-dominance as positive and 5′-dominance as negative is arbitrary, but has to be applied consistently.

### 1.2. CDA Considering the Purine/Pyrimidine Divide

Hence, so far simply considering nucleotides as letters enables to define CDA for 16 + 12 + 12 = 40 codons. The definition of codon side dominance for the remaining 24 codons requires considering physicochemical properties of the nucleotides forming the codons, because different types of nucleotides occupy each of the three codon positions. Nucleotides belong to purines (A, G) or pyrimidines (C,T/U). This chemical grouping can be used to define which nucleotide at 5′ or 3′ codon extremities differs most from the nucleotide at codon midposition. This approach defines CDA > 0 for eight among the 24 remaining codons: CTA, CTG, TCA, TCG, GAT, GAC, AGT, and AGC. Similarly, this defines CDA < 0 when the chemical type (purine/pyrimidine) of the first codon position differs from that at the two remaining positions, for codons ATC, GTC, ACT, GCT, TAG, CAG, TGA, and CGA.

### 1.3. CDA and Complementarity

Defining CDA for the eight remaining codons, ATG, GTA, ACG, GCA, TAC, CAT, TGC and CGT, requires adding a further criterion. Previously, the nucleotide at the 1st or 3rd codon position that is the complement of the nucleotide at codon midposition was considered most different from midposition, so that GTA, ACG, CAT and TGC have CDA > 0, and ATG, GCA TAC and CGT have CDA < 0.

### 1.4. CDA and Properties of the Translational Apparatus, the Genetic Code and Protein Translation

The CDA scores given to each codon according to the above rules [13] are shown in Table 1. Previous analyses showed that CDA averaged across all synonymous codons coding for a given amino acid is negative for eight among ten amino acids aminoacylated to their cognate tRNAs by class I tRNA synthetases (E, I, M, Q, R, V, W, Y); the two exceptions with CDA > 0 are Cys and Leu. Similarly, for eight among ten amino acids aminoacylated to their cognate tRNAs by class II tRNA synthetases, mean CDA across synonymous codons is positive: D, F, G, H, K, N, P, S, exceptions with CDA < 0 are Ala and Thr. Hence, CDA associates with tRNA synthetase classes for 16 among 20 (80%) amino acids [13]. Mean CDA for synonymous codons correlates negatively with secondary structure conformational indices of cognate amino acids, notably with the antiparallel beta sheet conformational index of amino acids (with a correlation r = −0.642) [13]. The latter result is particularly remarkable because it links codon and amino acid properties: properties of the symbol (codon) are related to those of the symbolized (amino acid).

In addition, mean CDA decreases with the order of inclusion of amino acids in the genetic code, according to the majorities of genetic code inclusion hypotheses as reviewed in [14], in agreement with genetic code evolutionary hypotheses that assume that class II tRNA synthetases and their cognate amino acids were first included in the genetic code [15,16]. This negative correlation is strongest (r = −0.775) with the genetic code inclusion order derived from the genetic code origin self-referential model [17] (presumed earliest to presumed latest amino acids: G, S, L, D, N, E, P, K, F, R, A, C, T, V, H, Q, I, M, Y and W).

### 1.5. Physicochemical Nucleotide Properties and Reference Codon Position

Here, the CDA approach is refined at two levels: 1. in relation to the physicochemical properties grouping nucleotides, previous analyses focused on the purine/pyrimidine divide between nucleotides; and 2. in relation to which nucleotide is considered as reference to compare with the remaining nucleotides, as previous analyses considered as reference codon midpositions. Hence the 64 codons are examined in relation to CDA as defined by the strength of codon-anticodon associations, along the weak/strong divide of complementary interactions (A:T vs. C:G), and according to tautomeric transformations nucleotides undergo, those with amino vs. imino isoforms (A, C) and keto vs. enol isoforms (G, T) [18,19]. Metal-binding promotes the rare nucleotide isoforms (enol and imino) [20].

In addition, the 64 codons are examined for CDA according to each of the three nucleotide properties (purine/pyrimidine; weak/strong complementary interactions, and tautomerisms) after considering as reference codon position the 1st and the 3rd positions, instead of the 2nd position as done previously [13]. This principle was also applied in other fields, for example, modernized classical genetics changed the referential system [21]. Here, the procedure implies that the swapped N2N1N3 codon codes for the same amino acid as the unswapped N1N2N3 codon: for example, N1N2N3 CTN which codes for Leu, still codes for Leu after N2N1N3 swap, resulting into codon N2N1N3 TCN. Similarly, N1N2N3 TCN (codes for Ser) codes for Ser after the N2N1N3 swap that produces the swapped codon N2N1N3 CTN.

## 2. Results

### 2.1. Purine-Pyrimidine-Based CDA Completed by Other Nucleotide Properties

The first CDA scores presented previously categorize 56 codons into CDA = 0, CDA < 0 and CDA > 0 based on the purine/pyrimidine grouping of nucleotides. CDA for the eight remaining codons was defined by complementarity between the nucleotide at 2nd codon position and either 1st or 3rd codon position nucleotides. This complementarity-based rule completes the CDA definition of all 64 codons. Here, analyses test whether replacing this complementarity-based rule by one among two different rules based on nucleotide groupings deduced from the strengths of codon-anticodon interactions (A:T vs. C:G) and tautomers (A, C vs. G, T) strengthens correlations between mean CDA and properties of the translational machinery, such as tRNA synthetase classes, antiparallel beta sheet indices and the genetic code inclusion order according to the self-referential model for genetic code evolution.

The eight codons whose CDA could not be defined according to the purine/pyrimidine divide are scored assuming that the nucleotide at the codon extremity that has the weakest interaction with its complementary nucleotide defines CDA. Hence ATG, ACG, TAC and TGC have CDA < 0 (A:T interactions are weaker than G:C interactions). Codons GTA, CAT, GCA and CGT have CDA > 0 along that rationale (Table 1).

This produces a negative mean CDA for codons coding for Cys, in line with the overall tendency for negative CDA for amino acids aminoacylated by class I tRNA synthetases. The sign of mean CDA for other synonymous codon classes remains unchanged. Hence, this new rule for completing CDA improves the correlation between CDA and tRNA synthetase class, with 17 among 20 (85%) amino acids fitting the observed patterns (from *p* = 0.00295 to *p* = 0.000644, one tailed sign test, according to a binomial distribution assuming equal chances for negative and positive mean CDA to match class I and II tRNA synthetases). These new CDA scores also produce stronger correlations with antiparallel beta sheet indices (from r = −0.642, one tailed *p* = 0.00114, to r = −0.694, one tailed *p* = 0.000344) and with the genetic code inclusion order according to the self-referential model (from r = −0.775, one tailed *p* = 0.00003, to r = −0.79, one tailed *p* = 0.000017). The combined *p* value for these three tests according to Fisher’s method for combining *p* values changes from *p* = 2.9 × 10^−8^ to 1.4 × 10^−9^.

No such effects strengthening all three associations with CDA are observed when considering as dominant strong, rather than weak, complementary nucleotide interactions. Similarly, defining CDA for these eight remaining codons according to the keto-enol vs. amino-imino divide, or vice versa, does not improve these correlations between CDA and each tRNA synthetase class, antiparallel beta sheet index, and the order of genetic code inclusion according to the self-referential model.

This definition of CDA is arbitrary: strong rather than weak interactions could have been chosen. However, using weak interactions in this context systematically improves all three previously described associations between CDA and amino acid properties; strong interactions systematically decrease them. This heuristic justification opens the way for deeper understandings of the principles underlying the genetic code’s structure.

### 2.2. CDA Based Solely on Interaction Strengths between Complementary Nucleotides

Here, differences between strengths of complementary interactions for 1st vs. 3rd codon position nucleotides are used to define CDA for all 64 codons. Averaging these CDA scores across synonymous codons does not produce notable correlations with amino acid physicochemical properties (for example molecular weight, polarity and hydrophobicity scales, and conformational indices) and tRNA synthetase classes. This CDA score based on the strengths of complementary nucleotide interactions is highly correlated with the genetic code inclusion order of amino acids derived from Juke’s neutral theory of genetic code origin (r = 0.977, two tailed *p* = 0.0) [22] and that derived from Dillon’s coevolution theory (r = 0.907, two tailed *p* = 0.00000004) [23]. In other words, CDA defined such that complementary interactions at first codon positions are stronger than those at last codon positions (CDA < 0) finds that CDA < 0 is for ancient amino acids and CDA > 0 for amino acids recently integrated in the genetic code. This reflects results from other hypotheses that assumed that the genetic code originated from strong codon-anticodon interactions (hypotheses 6 and 40 in [14]).

### 2.3. CDA Derived Solely from Tautomeric Groups

Similarly to the above section, CDA can be defined for all the 64 codons based on the belonging of nucleotides at first and third codon positions to either the nucleotides with the rare imino (A, C) vs. the rare enol isoform (G, T). Defining side dominance according to the latter type of tautomerism, codon ATG, as an example, has CDA > 0, and codon TTC has CDA < 0. This tautomer-based CDA does not produce notable correlations with any amino acid physicochemical properties, nor with any genetic code inclusion order hypothesis. It is worth noting that arbitrarily deciding to define CDA according to A and C, the nucleotides with the imino tautomers, rather than G and T, produces identical results, multiplied by −1.

### 2.4. CDA with Reference Nucleotide at 1st Instead of 2nd Codon Position

Previous CDA analyses consider which among the nucleotides at 5′ vs. 3′ codon extremities is most similar vs. differs most from the nucleotide at the codon’s midposition. In other words, CDA as defined previously considers the nucleotide at codon midposition as a reference.

An alternative CDA can be constructed with the first codon position as reference. The most practical method for this is to swap the nucleotides at 1st and 2nd codon positions. This could be tantamount, from an evolutionary perspective, to assuming that a more primitive genetic code existed where 1st and 2nd codon positions were swapped, while the codon was coding for the same amino acid. This would assume that codon structure evolved from N2N1N3 → N1N2N3. Each of the 64 swapped codons can be scored for CDA according to each of the nucleotide properties used for the regular N1N2N3 codons, that is the purine/pyrimidine, weak/strong complementary interactions and the keto-enol/amino-imino divides. Note that rational analyses based on 1st vs. 2nd codon position swaps do not necessarily imply that codon positions were physically swapped. These could also imply that differences between nucleotides at 2nd and 3rd codon positions as compared to 1st codon position nucleotides determined molecular interactions of codons with other molecules (for example amino acids, enzymes, anticodons), rather than differences between nucleotides at 1st and 3rd codon position compared to those at 2nd codon positions.

Mean CDA for synonymous codons according to each of these properties were calculated and correlations with amino acid physicochemical properties and genetic code inclusion hypotheses for the cognate amino acids coded by these codons were examined. There were no notable correlations between any of these variables and mean CDA for swapped codons, calculated according to each of the purine/pyrimidine and the weak/strong complementary interaction divides. Mean CDA for N2N1N3-swapped codons according to tautomeric isoforms (Table 2) produces notable correlations with two physicochemical and structural properties of amino acids. The tautomeric-based swapped CDA decreases with the Kyte and Doolittle amino acid hydropathicity [24], r = −0.702, two tailed *p* = 0.000556, and the tendency of amino acids to participate in parallel beta sheets [25], r = −0.753, two tailed *p* = 0.000127 (Figure 1a,b). Note that the decision to define this CDA according to G and T, rather than A and C, is arbitrary. Results with A and C would be identical but multiplied by −1. Correlation analyses also reflect this by using two-tailed statistical tests.

Amino acid tendency for parallel beta sheet formation is strongly correlated with the Kyte and Doolittle amino acid hydropathicity (r = 0.845, *p* < 0.0001). We calculated the multiple regression of CDA with both properties, accounting for collinearities between the conformational index and hydropathicity. This results in a multiple r = 0.751 (two tailed *p* = 0.001), with statistically significant negative slopes for each variable (parallel beta sheet formation, t = −4.442, *p* = 0.000357 and hydropathicity, t = −4.5, *p* = 0.000316, two tailed tests). Hence, CDA associates with both amino acid properties, also after accounting for colinearities between them.

These various results remain highly statistically significant also when applying the nonparametric Spearman rank correlation test.

Note that CDAs calculated after swaps between second and third positions producing the N1N3N2 codon structure do not produce notable associations with amino acid properties.

Codon usages in theoretical minimal RNA rings were previously examined according to CDA defined by the purine/pyrimidine divide [26]. Theoretical minimal RNA rings are theoretical constructs aimed at mimicking primordial RNAs [27,28], defined by the following constraints: the shortest sequence coding only once for each amino acid, a stop and a start codon, and forming a stem-loop hairpin conferring the RNA some protection against degradation. Exactly 25 circular RNAs with 22 nucleotides match these constraints after three translation rounds of partially overlapping codons. Interestingly, these theoretical RNA rings defined mainly by coding constraints surprisingly resemble tRNAs [29,30,31,32] and share several unexpected properties with protein coding sequences [33,34,35]. Among these 25 RNA rings, 23 are slightly biased for CDA > 0 when defining CDA according to the keto/imino divide (one more codon with positive than negative CDA). The two remaining RNA rings have equal numbers of codons with positive and negative CDA. This low variability in this type of CDA among RNA rings precludes further analyses.

## 3. Discussion

### 3.1. Improving CDA for Regular Codons

The concept of CDA enables to construct codon-specific scores based solely on the positions of nucleotides in each codon along very simple rationales based on palindromic codon structures and departures from symmetry. These scores, applied to the regular genetic code, along simple nucleotide properties such as the purine/pyrimidine divide, correlate with important features of the translational apparatus, such as the tRNA synthetase class that aminoacylates the codon’s cognate amino acid to its tRNA, the amino acid’s tendency to form antiparallel beta sheets, and the order of amino acid inclusion in the genetic code according to the self-referential model of genetic code evolution.

The purine/pyrimidine divide does not enable to define CDA for eight codons. Completing CDA for these remaining codons by applying onto them a second property, that of weak/strong complementary interactions, while considering that codon side dominance is defined by the nucleotide at the codon extremity that forms the weakest complementary interaction, strengthens correlations between CDA and each of the previously mentioned variables: tRNA synthetase class, the amino acid’s tendency to form antiparallel beta sheets, and the order of amino acid inclusion in the genetic code according to the self-referential model of genetic code evolution.

### 3.2. CDA for N2N1N3-Swapped Codons

For codons with the regular N1N2N3 structure, CDA defined by the keto-enol/amino-imino divide does not produce any notable correlations with amino acid properties. However, CDA had originally been defined by considering as reference the codon’s midposition. This is somewhat arbitrary, and CDA can be redefined while considering other codon positions as reference to define CDA. Considering the first codon position as reference, CDA scores can easily be constructed by swapping the positions of the 1st and the 2nd codon nucleotides, then proceeding as for regular, unswapped codons. This procedure could reflect an evolutionary swap between codon positions, indicating that the modern N1N2N3 codon structure we know could have been preceded by a more ancient N2N1N3 codon structure. Note that CDAs based on the N1N3N2 codon structure do not produce any notable associations with amino acid properties.

The correlations observed between amino acid properties and N2N1N3-CDA could bear witness to this evolutionary swap. Alternatively, it could reflect unknown processes affecting protein translation that relate to a hierarchy between nucleotides at codon positions that differs from what is known, where the midposition is most important, and the first codon position has intermediary coding importance. In the case of N2N1N3 codon structures, the first codon position would be the reference in relation to crucial amino acid properties such as polarity and the parallel beta sheet conformational index.

### 3.3. Ruhmer’s Transformation and Genetic Code Symmetry

The correlations between N2N1N3 codon structure according to the tautomer-defined CDA described here reflect a deeper principle of the genetic code’s structure. Indeed, here, CDA < 0 occurs for a synonymous codon family when N2 is either G or T, and CDA > 0 when N2 is either A or C. This corresponds to the symmetries in the genetic code related to Ruhmer’s transformation [36], where nucleotides are transformed along the rule T,C,A,G > G,A,C,T. The suggestion that this reflects an ancient code based on stereochemical codon-amino acid interactions [37] and regulation of protein synthesis [38] are in line with observations reported here on associations between CDA and amino acid properties.

## 4. Conclusions

Associations between amino acid properties and N2N1N3-CDA occur when constructing CDA along the keto-enol/amino-imino tautomeric isoform divide. In other terms, the type of tautomery for nucleotides at the different codon positions is informative in relation to the formation of parallel beta sheets and amino acid polarity/hydropathicity. It is possible that conditions in the primitive, hadean earth promoted keto → enol and amino → imino tautomeric isoforms much more than in current conditions under cellular protection. Hence, these chemical within-nucleotide changes would have had much stronger impacts on codon structures and a primitive translation system than in current conditions, explaining the correlations observed between swapped codon CDA and major physicochemical and structural properties of amino acids. The reasons for this remain unknown but statistical observations are strong and warrant further careful considerations and probably relate to Ruhmer’s symmetries.

It is worth noting that CDA is the only codon property that correlates with conformational secondary structure preferences of amino acids, CDA based on the purine/pyrimidine divide with antiparallel beta sheet formation, and CDA based on the keto/amino divide with parallel beta sheet formation. This suggests the potential use of CDAs in protein structure predictions.

## Figures and Tables

**Figure 1 ijms-21-00347-f001:**
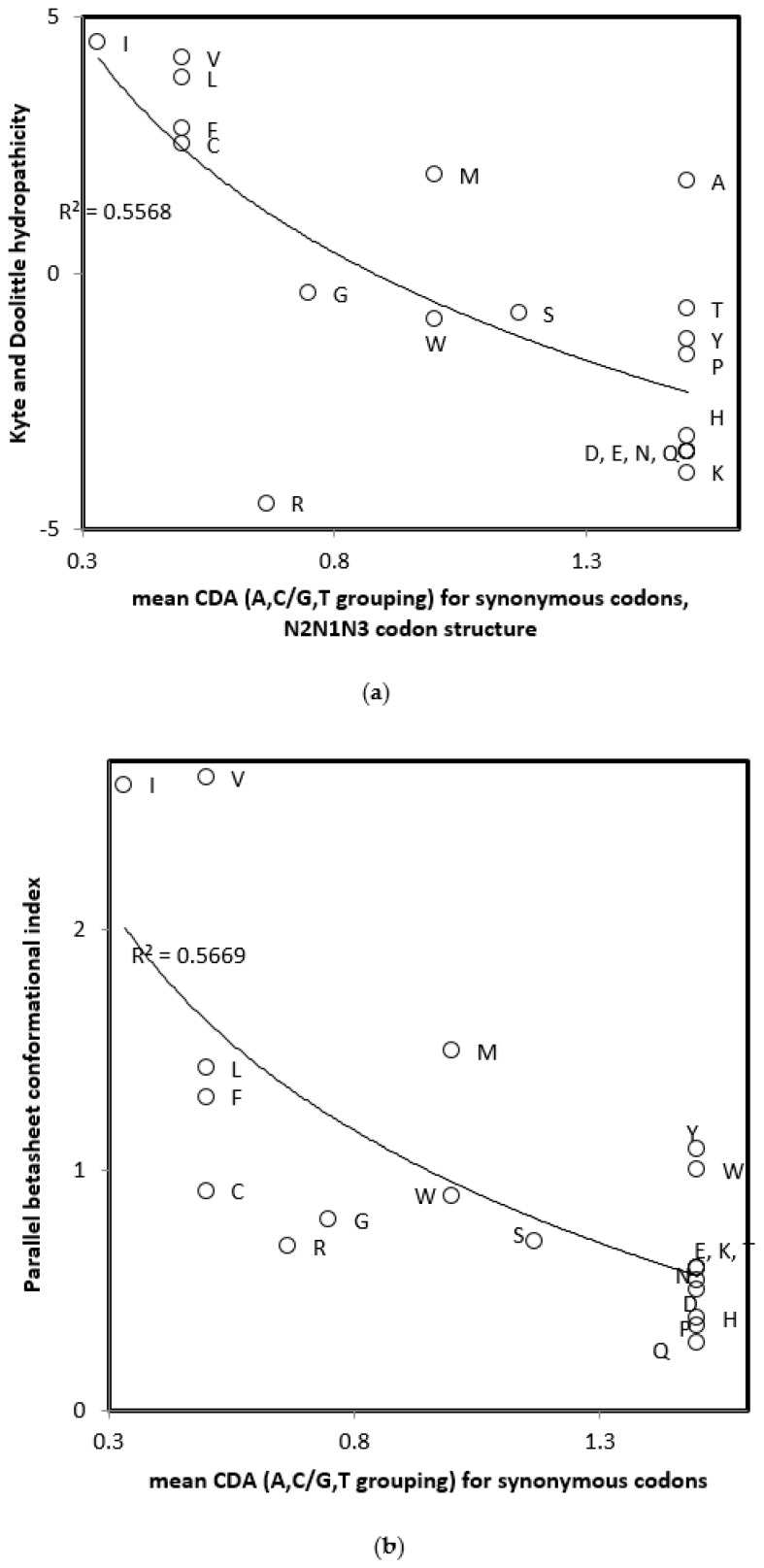
Correlations of amino acid physicochemical and structural properties with mean CDA for synonymous codons according to swapped N2N1N3 codon structure and tautomeric isoforms, from Table 2. (**a**) Kyte and Doolittle hydropathicity; and (**b**) parallel beta sheet conformational index.

**Table 1 ijms-21-00347-t001:**
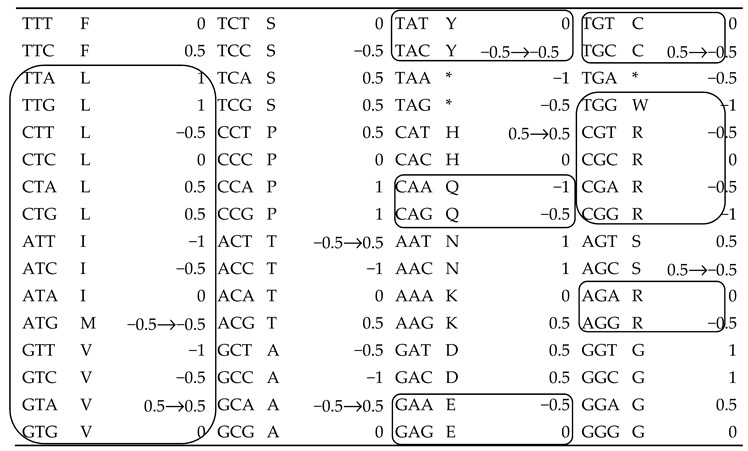
The codon directional asymmetry (CDA) of the genetic code’s 64 codons as defined by the purine/pyrimidine divide. Shaded nucleotides: the nucleotide most different from nucleotides at other positions, which determines the codon dominant side. CDA < 0 if the first (5′) codon position is most different, and CDA > 0 if the third (3′) position is most different. Frames indicate codons assigned to amino acids aminoacylated by class I tRNA synthetases. CDA of codons ATG, GTA, ACG, GCA, TAC, CAT, TGC and AGC was originally defined by complementarity of nucleotides at extreme codon positions and the midposition nucleotide because the purine/pyrimidine divide is uninformative for these codons regarding CDA. Assuming for these eight codons that codon side dominance is defined by the extreme (first or last position) nucleotide with the weakest complementary interaction defines CDA for these eight codons in this new analysis (* is for STOP).

**Table 2 ijms-21-00347-t002:** The tautomeric codon directional asymmetry (CDA) of the genetic code’s 64 codons as defined by the keto-enol/amino-imino divide, for N2N1N3 swapped codons. Side dominance is defined by occupancy of nucleotides with keto and enol tautomeric isoforms, G and T (* for STOP).

TTT	F	0	CTT	S	1	ATT	Y	1	GTT	C	0
TTC	F	−1	CTC	S	0	ATC	Y	0	GTC	C	−1
TTA	L	−1	CTA	S	0	ATA	*	0	GTA	*	−1
TTG	L	0	CTG	S	1	ATG	*	1	GTG	W	0
TCT	L	0	CCT	P	1	ACT	H	1	GCT	R	0
TCC	L	−1	CCC	P	0	ACC	H	0	GCC	R	−1
TCA	L	−1	CCA	P	0	ACA	Q	0	GCA	R	−1
TCG	L	0	CCG	P	1	ACG	Q	1	GCG	R	0
TAT	I	0	CAT	T	1	AAT	N	1	GAT	S	0
TAC	I	−1	CAC	T	0	AAC	N	0	GAC	S	−1
TAA	I	−1	CAA	T	0	AAA	K	0	GAA	R	0
TAG	M	0	CAG	T	1	AAG	K	1	GAG	R	0
TGT	V	0	CGT	A	1	AGT	D	1	GGT	G	0
TGC	V	−1	CGC	A	0	AGC	D	0	GGC	G	−1
TGA	V	−1	CGA	A	0	AGA	E	0	GGA	G	0
TGG	V	0	CGG	A	1	AGG	E	1	GGG	G	0

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
