# Peer review of "Codon Directional Asymmetry Suggests Swapped Prebiotic 1st and 2nd Codon Positions"

_ijms, 2020, doi:10.3390/ijms21010347_

Round 1

Reviewer 1 Report

The principle and limitation of author’s approach is well explained in their revised manuscript and the response from the authors.

Now I can agree that this study is a possible approach.

I recommend acceptance.

Reviewer 2 Report

Authors addressed all the comments of the reviewers in a satisfactory way.

This manuscript is a resubmission of an earlier submission. The following is a list of the peer review reports and author responses from that submission.

Round 1

Reviewer 1 Report

Seligmann and Demongeot addressed the comments previously raised by reviewers and improved the presentation of the results. Overall, the study merits publication. Just a couple of minor modifications to clarify the text:

Abstract line 20: dont -> don't

In general it is better to use 3rd than 3d to refer to third position.

293-297: there is an open parenthesis without the closing parenthesis.

Reviewer 2 Report

Authors previously defined Codon Directional Asymmetry (CDA)(Seligmann & Warthi, Comput Struct Biotechnol J 2017, 15, 412-424).

CDA was determined as following:

(1) Palindrome (CDA = 0)

(2) Identity to 2nd codon position (CDA for XXY >0, CDA for XYY <0)

(3) Identity of chemical group (purines/pyrimidines) to 2nd codon position (CDA for PrPrPu or PuPuPr >0, CDA for PrPuPu or PuPrPr <0)

(4) Complementation to 2nd codon position (ex. CDA for GTA >0, CDA for ATG <0)

In the previous paper, authors described that this CDA correlated with cognate amino acid tRNA synthetase classes, antiparallel betasheet conformation index and the evolutionary order defined by the self-referential genetic code evolution model.

In the present paper,

(4)’ Nucleotide groupings deduced from the strengths of codon-anticodon interaction [ex. CDA for TAC >0, which was evaluated <0 by (4)]

was newly defined, and a modified CDA was calculated.

The new rule improved the correlation between CDA and tRNA synthetase class, antiparallel betasheet indices and the genetic code inclusion order.

In addition,

(0) Use the first codon position as reference.

(4)’’ Nucleotide groupings deduced from the tautomeric isoforms.

were also defined, and another modified CDA was calculated.

The other modified CDA correlated with antiparallel betasheet conformation index and Doolittle’s hydropathicities.

Unfortunately, these newly defined rules includes fundamental problems as described below.

Major points:

In the newly defined rules [(4)’ and (4)’’], no reference nucleotide (2nd or 1st codon position) was used.

Therefore, plus or minus of CDA could be arbitrarily defined, suggesting that the correlation with biologically meaningful indices, such as antiparallel betasheet conformation index, can be improved by convenient definition of the new rules.

The newly defined rules may not be orthogonal from higher rules.

For instance, both CDAs for AAG and GGA are evaluated as 0.5 by the rule (2).

On the other hand, the CDAs can be also estimated using the rule (4)’.

In this case, CDA should be <0 for AAG, and >0 for GGA.

Minor points:

Abstract

Background, Methods and Results are confused each other.

Line 42:

(CDA, -> (CDA)

Results

2.2 is a copy of 2.1.

Line186 & 200

2.3 is duplicated.

//